# Interval Type-2 Fuzzy Logic Control-Based Frequency Control of Hybrid Power System Using DMGS of PI Controller

**Swati Rawat [1], Bhola Jha [1],\*, Manoj Kumar Panda [2] and Jyotshana Kanti [3]**

1. Department of Electrical Engineering, G. B. Pant Institute of Engineering and Technology, Pauri 246194, India; rawat.swatieee@gmail.com
2. State Project Implementation Unit, Dehradun 248015, India; pandagbpec@gmail.com
3. Department of Research and Development, Robotronix Engineering Tech Pvt. Ltd., Indore 451010, India; jyotshanakanti@gmail.com

\* Correspondence: bholajhaeee@gmail.com

**Abstract:** The load frequency control of a microgrid is one of the emerging areas due to the changes in demand and supply in power system. So the controllers' implementation must be changed accordingly. This paper proposes an interval type-2 fuzzy logic-based, dual-mode gain scheduling (DMGS) of the proportional and integral controller in which the gains of the PI controller werescheduled through the dynamic selector. This proposed controller was implemented ina hybrid microgrid power system in which nonconventional energy sources wereadded to each area of the conventional power plant, which madethe system much more prone to frequency variations. The controller was designed for three areas, consisting of a photovoltaic (PV) system, a wind power system, a fuel cell and a diesel engine/hydropower generator in which the generation rate constraint (GRC) was considered as a nonlinearity. The proposed power system was investigated under various load conditions in the MATLAB/SIMULINK environment. A comparative evaluation of changes in frequency, tie-line power fluctuations and variations in area control errors for the test system showed the effectiveness of the current approach over simple fuzzy PI and a conventional PI-controlling approach.

**Keywords:** renewable power; interval type-2 fuzzy logic control (IT2FLC); power consumption for transactions; dual-mode gain scheduling (DMGS); load frequency control (LFC); area control error (ACE); tie-line power fluctuations

## 1. Introduction

Our world is dependent on conventional energy sources, but these resources may vanish after some decades. In such situations, nonconventional resources are the choices that can provide sustainable and environmentally friendly electricity. Renewable or nonconventional power systems are very beneficial, but the main problem is the instability of voltage and frequency. In this paper, work has been presented to analyze frequency variations in interconnected test systems thatconsist of renewable energy resources. When generation and load demand differ from each other, load-frequency fluctuation occurs. To avoid frequency instability, there are several techniques reported as per research papers and literature. It depends upon the designer to choose a typical controller thatmay be suitable for the test system to avoid frequency-variation problems.

A new technique of continuous under-frequency load shedding (UFLS) has been found for frequency control [1]. A closed-form solution of frequency dynamics was analyzed with the UFLS scheme. A distributed economic model predictive control (EMPC) was proposed for economic load dispatch and frequency-variation problems in multi-area power systems [2]. Asymptotic stability was proven using the EMPC algorithm. A frequency-variation control technique was also presented to give a trustworthy count governor and generator inertia response [3]. This scheme also played a lead role to improve inter-area oscillation damping. To avoid load frequency variation in wind farms, using

PID controllers has been also proposed in the literature [4]. Parameters of this controller were optimized using particle-swarm optimization. The latest controlling technique has been presented, using a fractional-order fuzzy PD and I controller to avoid frequency variations in a microgrid of a ship power system [5]. For tuning parameters of the controller, a black hole optimization algorithm was used. An augmented load frequency control was implemented in [6] to ensure the frequency regulation under dirunal conditions in renewable-rich power systems. The distributed automatic load control for local measurement and local communication was proposed in [7] for restoring nominal frequency and scheduled tie-line power flows. Peturbation-estimation-based load frequency control was proposed in [8]. Here, peturbation implied the measurement error or time delay. The smart transformer (ST) working on the load sensitivitywas used in [9] for the primary frequency regulation by shaping the load consumption. The robust sliding-mode control strategy was proposed in [10] against frequency deviations caused by power unbalances or time delays in multiarea interconnected microgrid power systems. An attempt was made to apply the cascaded PI-PD controller in [11] for hybrid power systems in which the gains of the controller's parameter were optimized by the particle swarm optimization-gravitational search algorithm (PSO-GSA) method. An LFC scheme using a simple fuzzy logic controller was also proposed for a renewable hybrid power system with energy storage (fuel cell system) [12]. Research work for an LFC scheme using a type-2 fuzzy logic PID controller for hybrid isolated power systemswasalso presented in [13]. This technique also improved performance index parameters of the transient response characteristics. This analysis was only for a single-area power system with renewable sources. There is an emerging technology and applications related to the internet of things (IOT) presented in [14]. It can be utilized in smart grids to control various parameters. This research work proposed applications for fundamental performance limits for the high-reliability and low-latency wireless internet of things. Another technology, which is called visible light communication function, can help to operate the intelligent power grid system presented in [15].

Controllers' implementation must be change according to the changes in the behaviour of power system. This paper proposes a hybrid microgrid power system in which nonconventional energy sources wereadded to each area of the conventional power plantin all the interconnected areas, which makes the system much more prone to frequency variations. In such a system, a conventional controller, such as a PID FLC, optimization-based controller may not suffice. Toa certain extent as reported, the optimization-based controller was successful, but one of the drawbacks was a long converging time. Therefore, the design of a robust intelligent controller was required for the minimization of frequency error. In this way, this paper presents an interval type-2 fuzzy logic-based, dual-mode gain scheduling (DMGS) of the proportional and integral controller in which the gains of PI controller were scheduled through the dynamic selector switch instead of the optimization algorithm. The proposed technique was a combination of IT2FL, PI controller and dynamic selector switch; hence, the technique is said to be a hybrid controller-based technique.

This paper is structured as described: Section 2 provides the system description and modeling of the proposed automatic generation control (AGC) model considering RES power fluctuation. Section 3 presents the application of IT2FLC in a proposed interconnected system. Section 4 describes the results and discussion. Conclusions are drawn in Section 5.

## 2. System Description and Modeling of Proposed Nonlinear Interconnected Power System

The proposed system under study consisted of three areas. All three areas were hybrid systems. Each area consisted of two RESs (WTG and PV system) and a conventional source (DEG/hydro system). Figure 1 represents this multiarea hybrid model in detail. Generation Rate Constraint (GRC), a nonlinearity that occurs in power systems due to the rate of change in power systems, wasalso considered in this test system. The main reason for this nonlinearity was thermal and mechanical movements in the power system. Table 1 represents data for the proposed system.

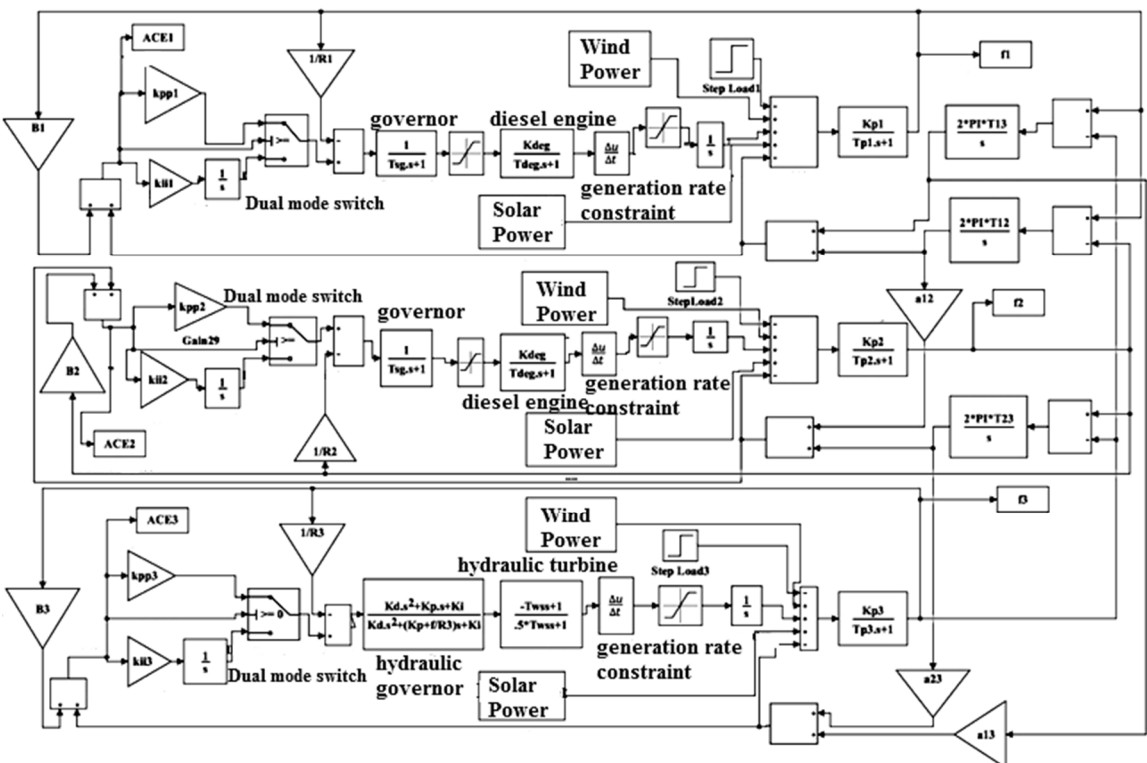

**Figure 1.** Three-area AGC model considering RES.

**Table 1.** Data for interconnected, three-area power system considering RES.

| Parameters | Value |
|---|---|
| Rating of each area | 0.1 MW |
| Base power | 0.1 MW |
| $K_{P1}$, $K_{P2}$, $K_{P3}$ | 120 Hz/p.u. MW |
| $T_{P1}$, $T_{P2}$, $T_{P3}$ | 20 sec |
| $T_{12}$, $T_{13}$, $T_{23}$ | 0.545 p.u. |
| $a_{12}$, $a_{31}$, $a_{23}$ | −1 |
| $R_1$, R2, $R_3$ | 2.4 Hz/p.u. MW |
| $T_W$ | 1 sec |
| $K_d$ | 4.0 |
| $K_i$ | 5.0 |
| $K_P$ | 1.0 |
| F | 50 Hz |
| Base Power | 0.1 MW |

### 2.1. Wind Turbine Model

A basic equation relating to wind speed and the mechanical power generated by awind turbine is as follows [16]:

$$P_o = C_{pc}(\lambda, \beta)\frac{1}{2}\rho A V_w^3 \tag{1}$$

where $P_0$ is the output mechanical power of the turbine (W), $\lambda$ is the tip speed ratio, $\beta$ is the blade pitch angle (°), $C_{Pc}$ is the performance coefficient of the turbine, $\rho$ is air density (kg $(m^3)^{-1}$), $V_w$ is wind speed ($ms^{-1}$) and $A$ is turbine swept area ($m^2$)

Normalization and simplification can be done in Equation (1) for particular values of $\rho$ and $A$. The new equation of the system in per unit (p.u.) is as follows:

$$P_{o\text{-}pu} = k_p C_{Pc\text{-}pu} V_{w\text{-}pu}^3 \tag{2}$$

where $P_{o\text{-}Pu}$ is the nominal power for particular values of $\rho$ and $A$ (p.u.), $K_p$ is power gain for $C_{Pc\text{-}pu} = 1$ and $V_{w\text{-}pu} = 1$ p.u. ($k_p \leq 1$), $C_{Pc\text{-}pu}$ is per unit (p.u.) value of the performance coefficient $C_{Pc}$ and $V_{w\text{-}pu}$ is the p.u. value of base wind speed.

Figure 2 shows a modified Simulink model of the wind turbine in which wind speed was the input and mechanical torque was the output that was applied to the induction generator. Corresponding to various values of wind speed, turbine speed and turbine output power characteristics are shown in Figure 3. Turbine output power remained zero from 0–0.2 p.u. turbine speed due to inertia in the turbine.

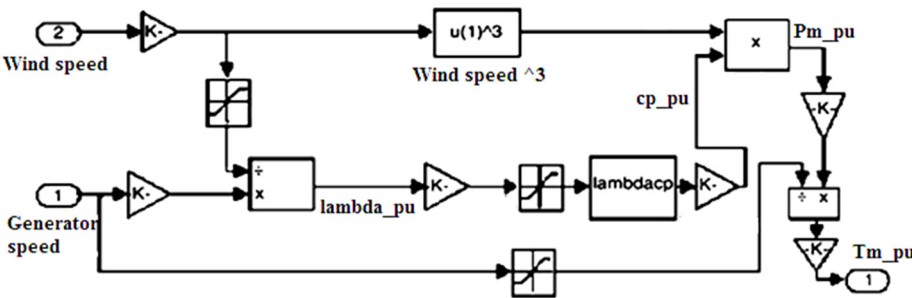

**Figure 2.** Simulink wind turbine model.

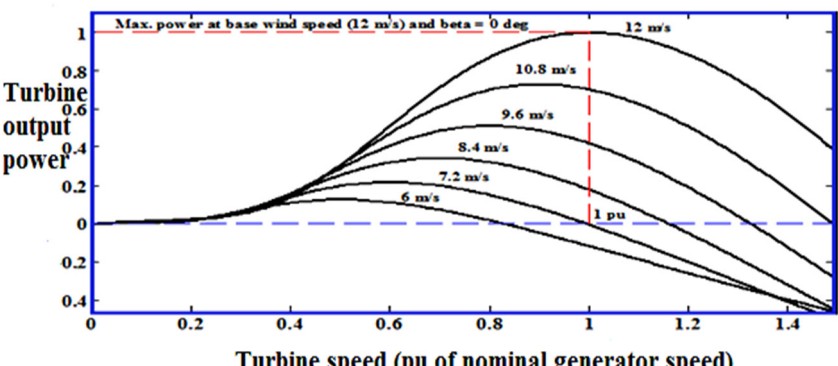

**Figure 3.** Turbine power characteristics.

### 2.2. Photovoltaic Model

The solar cell was a semiconductor diode. It converts light energy into electrical energy, which is called the photovoltaic effect. Figure 4 represents a solar cell.

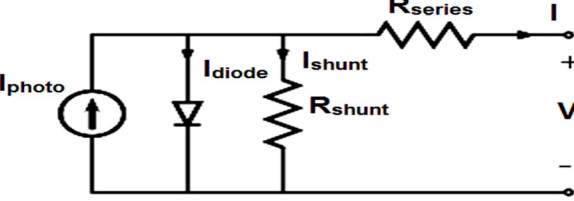

**Figure 4.** Circuit diagram of solar cell.

The equation of the ideal PV cell is given as below [17]:

$$I_{cell} = I_{photo} - I_{sat}\left[\exp\left(\frac{V}{AV_{th}}\right) - 1\right] \tag{3}$$

where $I_{photo}$ is photocurrent (A), $I_{sat}$ is reverse saturation current (A), $V$ is diode voltage (V), $V_{th}$ is thermal voltage ($V_{th} = 27.5$ mV at 25 °C) and $A$ is the ideality factor of the diode.

A single solar cell provides approximately 0.5 V. Cells are connected in series combination for high voltage and parallel-connected for high current to form a PV module for desired voltage and current. The PV module voltage-current (VI) characteristic equation is mentioned here under.

$$I_{module} = N_p I_{photo} - N_p I_{sat} \left[ \exp \left( \frac{V_{module}/N_s + I_{module}/N_s}{AV_{th}} \right) \right] - \frac{(N_p/N_s)V_{module} + I_{module}R_{series}}{R_{shunt}} \tag{4}$$

where the cell's parallel and series number is $N_p$ and $N_s$.

Figures 5 and 6 show VI and PV characteristics, respectively, for PV modules.

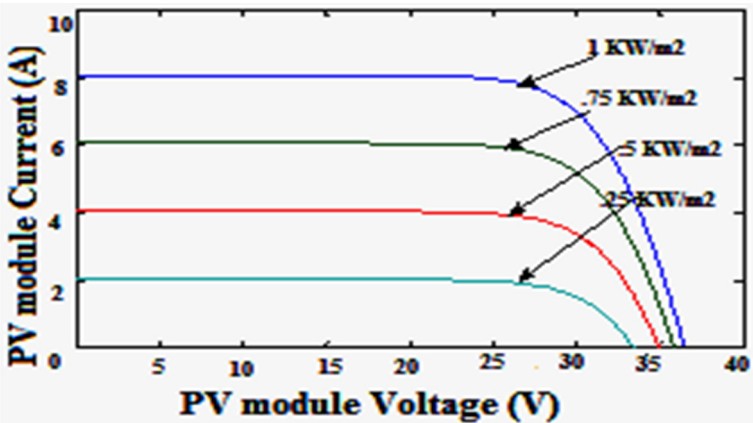

**Figure 5.** V-I characteristic of PV module [17].

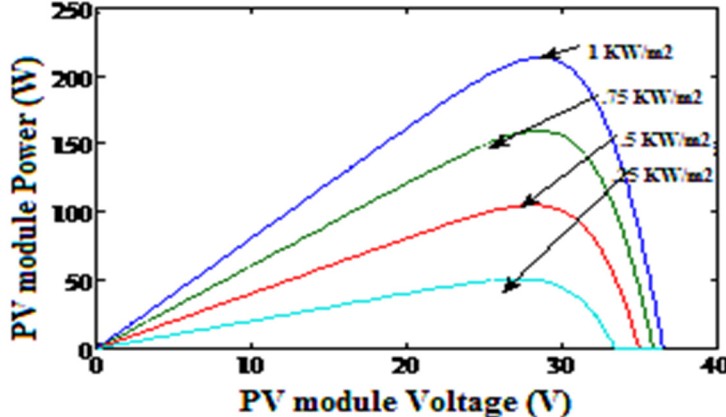

**Figure 6.** P-V characteristics of PV module [17].

The SESG-ND-216u1F solar module has used for the modeling of PV power system [18]. The model was evaluated in the embedded MATLAB function and solved by the Newton–Raphson method.

### 2.3. Diesel Engine Generator (DEG) Model

A speed governor and diesel engine are the two main components of DEG models. There is a standard block-diagram model for DEGs as can be seen in Figure 7 [19].

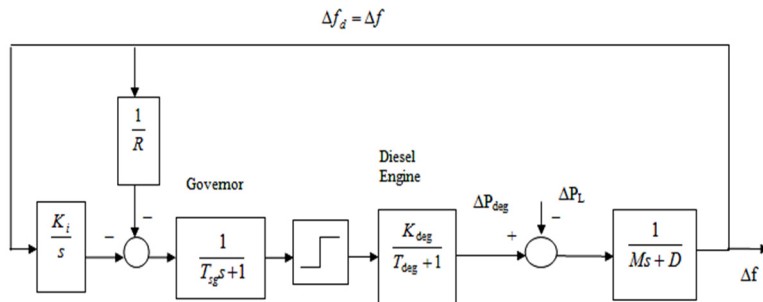

**Figure 7.** The standard model for DEGs.

The transfer function of the DEG model is as follows:

$$P_{diesel} = -\left(\frac{1}{s_r} + \frac{I}{s}\right)\left(\frac{1}{Ts+1}\right)\left(\frac{k}{T_d+1}\right)\Delta f \tag{5}$$

where $P_{diesel}$ = DEG power, $I$ = integral gain constant, $T$ = speed governor time constant, $K$ = diesel engine gain constant, $S_r$ = speed regulation and $T_d$ = diesel engine time constant.

### 2.4. Hydro Model

The transfer function of the hydraulic turbine is represented as follows [20]:

$$\frac{-t_{water}s + 1}{0.5t_{water}s + 1} \tag{6}$$

where $t_{water}$ = water starting time.

The transfer function of the hydraulic governor is:

$$\frac{K_d s^2 + K_p s + K_i}{K_d s^2 + (K_p + \frac{f}{R})s + K_i} \tag{7}$$

## 3. IT2FLS and Its Application in the Proposed Test System

### 3.1. IT2FLSs and Interval Type-2 Fuzzy Logic Controller

The vagueness in rules can be handled by an IT2FLS. There is uncertainty or fuzziness in the membership functions of an IT2FLS that is not certain or definite. A footprint of uncertainty (F) in IT2FLSsisshown in Figure 8 [21].

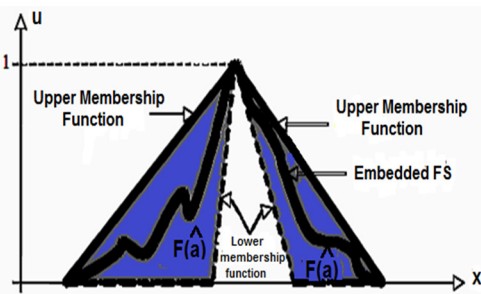

**Figure 8.** The footprint of uncertainty FOU for IT2FSs.

The Shaded portion represent FOU, Lower Membership Function (LMF) represented by dashed line, Upper Membership Function (UMF) represented by solid line and an embedded FS by Wavy line.

The bounded area between the lower and uppermembership functions is called F. It gives a further degree of autonomy. Because of this reason, a simple or conventional fuzzy

logic system with an identical number of membership functions cannotconquer output likea type-2 FLS can conquer. $\hat{a}$ type-2 fuzzy logic set is mentioned as follows:

$$\hat{a} = \int_{y \in Y} \int_{u \in J_y \subseteq [0,1]} \frac{1}{(y,u)} = \int_{y \in X} [\int_{u \in J_y \subseteq [0,1]} \frac{1}{u}]/y \tag{8}$$

where $y \in Y$ and $u \in U$ represent the primary and secondary variables. $J_Y$ is the primary membership function of $y$. The value of secondary grades of $\hat{a}$ is unity. The vagueness $\hat{a}$ is presented by the union of primary membership functions. The larger the amount of uncertainty, the larger the footprint of uncertainty (FOU) will be. UMF and LMF are denoted as $\overline{\mu}a(x)$ and $\underline{\mu}a(x)$.

$$\overline{\mu \hat{a}}(x) = \overline{F(\hat{a})} \tag{9}$$

$$\underline{\mu \hat{a}}(x) = \underline{F(\hat{a})} \tag{10}$$

LMF and UMF $\hat{a}$ cover a footprint of uncertainty. Both membership functions are kinds of type-1 membership functions. Figure 9 represents the block diagram of IT2FLC [22]. FOU can be represented as:

$$F(\hat{a}) = \cup_{\forall y \in Y} J_y = \{(y,u) : u \in J_y \subseteq [0,1]\} \tag{11}$$

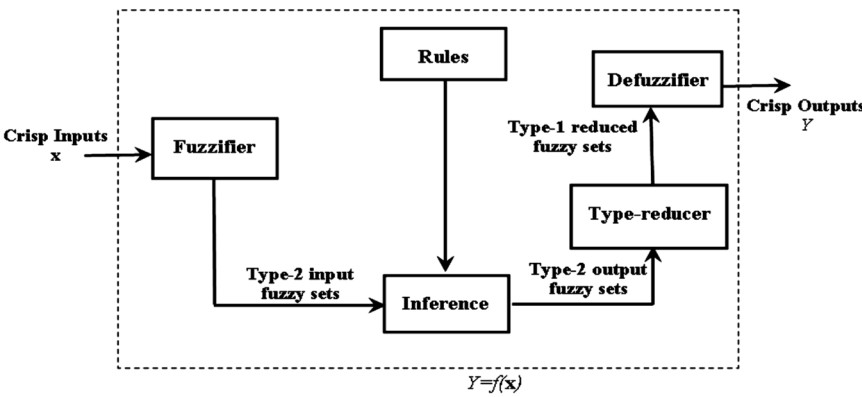

**Figure 9.** Interval type-2 fuzzy logic controller.

Like a conventional fuzzy system, IT2FS is also based on a rule-based system. It consists of five operations, i.e., fuzzifier, rules, inference, type reducer and defuzzifier. Fuzzy values can be obtained from a fuzzifier. It converts actual input data into fuzzy sets. The antecedents and consequents relationship is a set of rules in the rule base. These rules are combined in inference. A type reducer converts the type-2 fuzzy set into the simple fuzzy set. The work of a defuzzifier is to convert fuzzy sets into actual or crisp signals.

### 3.2. Type-2 Fuzzy Logic-Based Gain Scheduling of PI Controller for Proposed Test System

There is a technique, called gain scheduling, which is frequently used in nonlinear systems. As per the change in system dynamics, the parameters also change quickly. Parameter estimation is not required in gain scheduling. As compared to automatic tuning in controllers, it is the easiest way. The DMGS technique for PI controllers has also been presented in the literature [23]. Figure 10 represents the block diagram of the proposed technique for a PI controller.

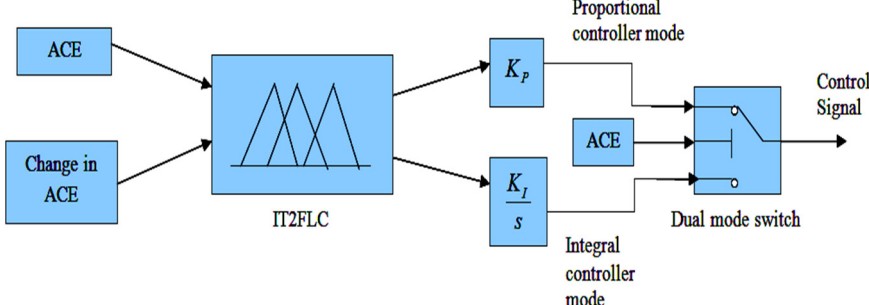

**Figure 10.** Block diagram of IT2FLC-based DMGS of PI controller.

Membership functions of $K_P$ and $K_I$ are shown in Figures 11 and 12.

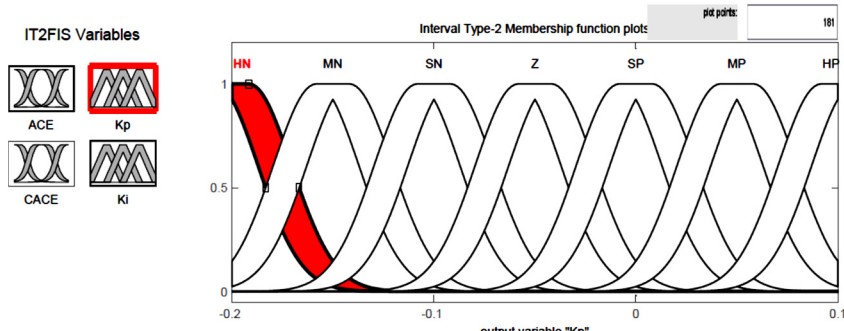

**Figure 11.** IT2FLC membership function of $K_P$.

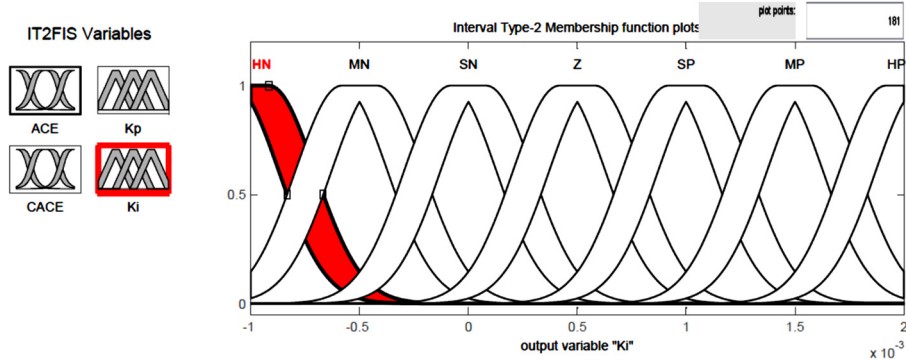

**Figure 12.** IT2FLC membership function of $K_I$.

Figures 13 and 14 show output surface viewers for $K_P$ and $K_I$.

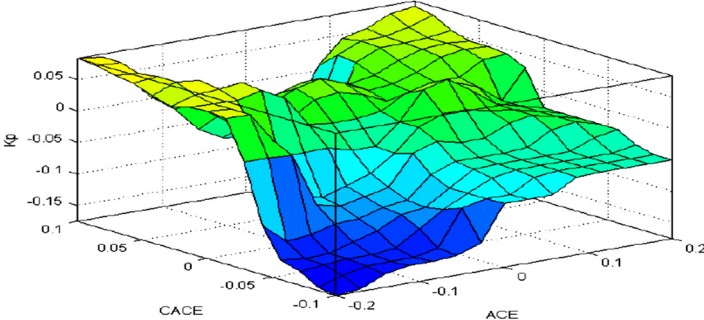

**Figure 13.** MATLAB surface view of $K_P$.

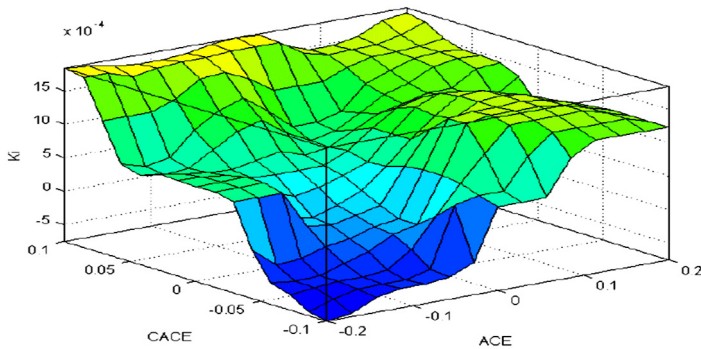

**Figure 14.** MATLAB surface view of $K_I$.

In this paper, an intelligent IT2FLC technique was utilized to set the parameters of the proportional and integral controller according to ACE and change in ACE. The dual-mode switch was used to connect between integral mode and proportional mode based upon the control signal. It is a dynamic selector switch for both parameters of the PI controller. In a steady-state condition, ACE for area 1 should be zero if the fluctuation in frequency and tie-line power are zero.

## 4. Results

To exemplify the act of the IT2FLC-based DMGS of the PI controller, simulations of the test model were considered for four conditions as follows.

### 4.1. Condition 1

Figures 15–17 represent the frequency variation, inter-tie-line power change and ACE, respectively, for the three different areas due to a 0.1 per unit (0.001 MW) change in step load demand in area 1.

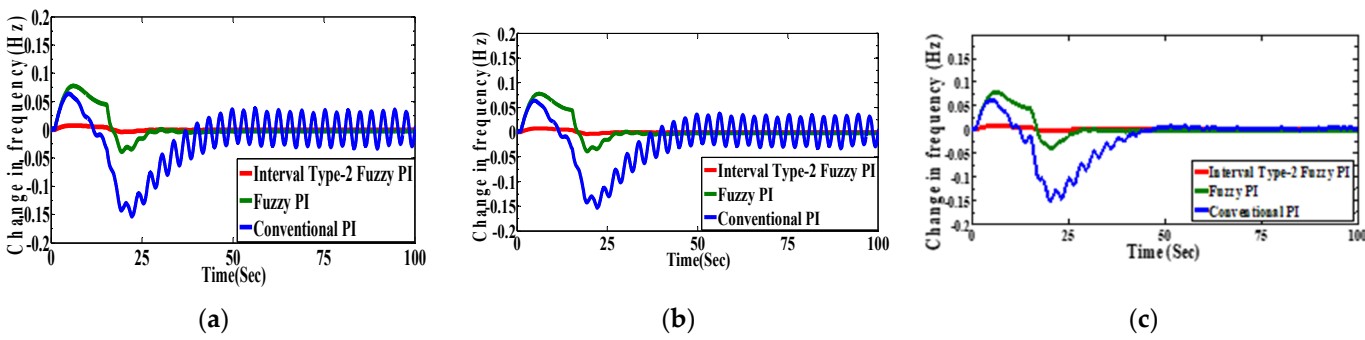

**Figure 15.** Frequency variations of (**a**) area 1, (**b**) area 2 and (**c**) area 3 using IT2FLC fuzzy-PI, fuzzy-PI and PI controller due to step load change of 0.1 p.u. in area 1.

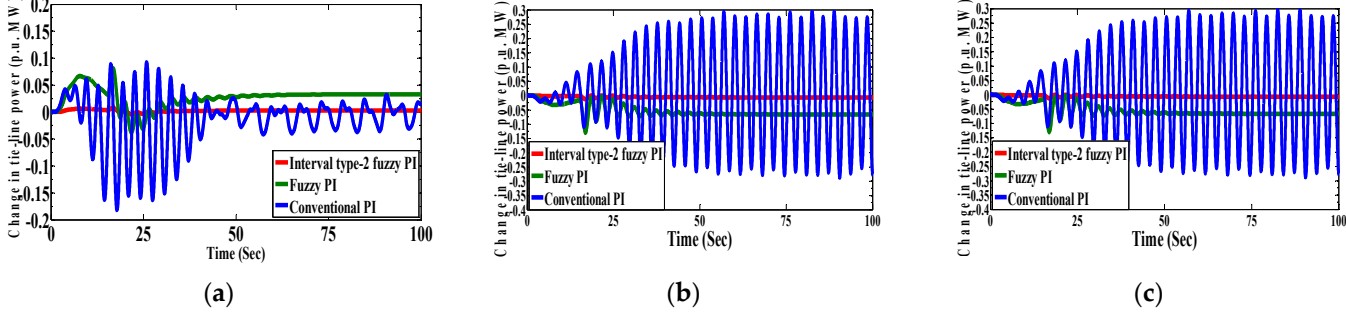

**Figure 16.** Tie-line power variations of (**a**) area 1, (**b**) area 2 and (**c**) area 3 using IT2FLC fuzzy-PI, fuzzy-PI and PI controller due to step load change of 0.1 p.u. in area 1.

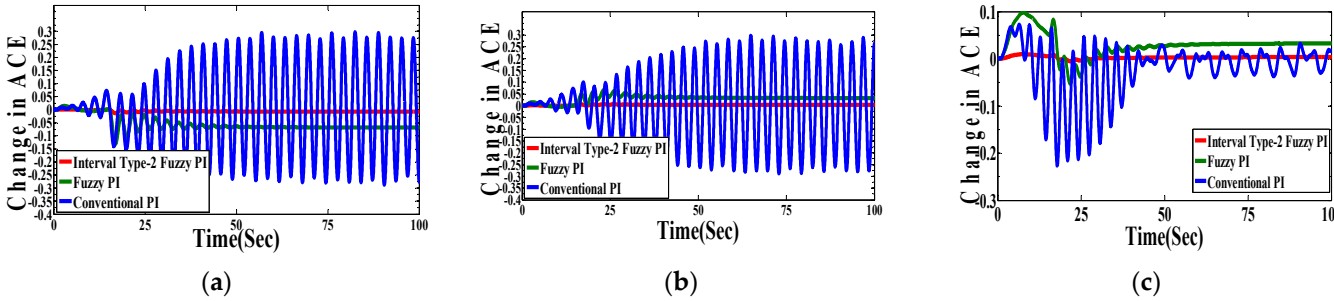

**Figure 17.** ACE variations of (**a**) area 1, (**b**) area 2 and (**c**) area 3 using IT2FLC fuzzy-PI, fuzzy-PI and PI controller due to step load change of 0.1 p.u. in area 1.

*4.2. Condition 2*

Figures 18–20 show the frequency variation, inter-tie-line power change and ACE, respectively, for the three different areas due to a 0.1 per unit (0.001 MW) change in step load demand in area 2.

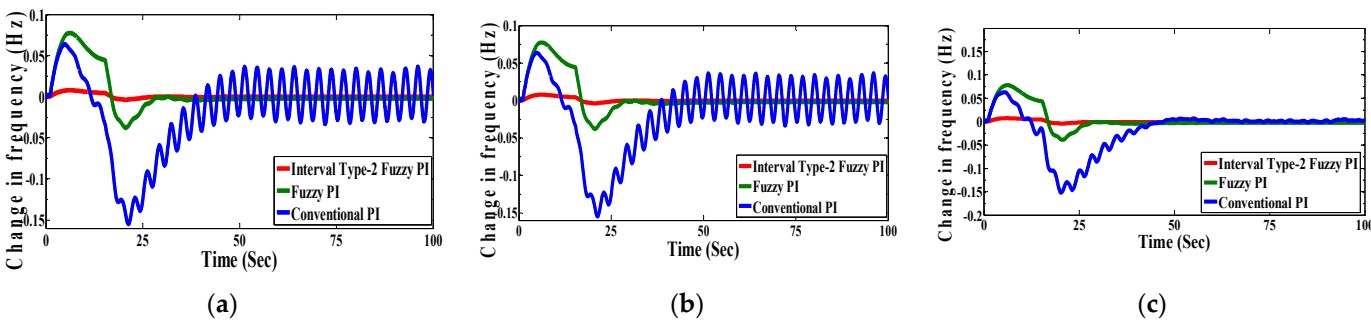

**Figure 18.** Frequency variations of (**a**) area 1, (**b**) area 2 and (**c**) area 3 using IT2FLC fuzzy-PI, fuzzy-PI and PI controller due to step load change of 0.1 p.u. in area 2.

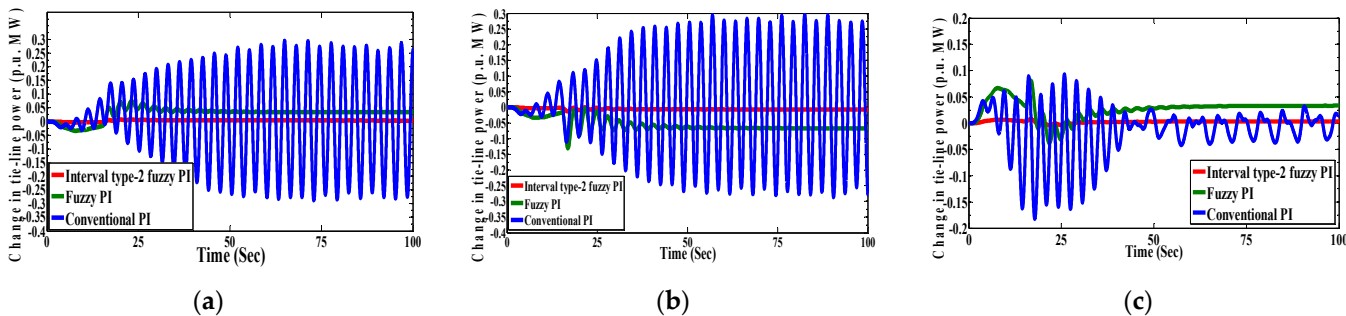

**Figure 19.** Tie-line power variations of (**a**) area 1, (**b**) area 2 and (**c**) area 3 using IT2FLC fuzzy-PI, fuzzy-PI and PI controller due to step load change of 0.1 p.u. in area 2.

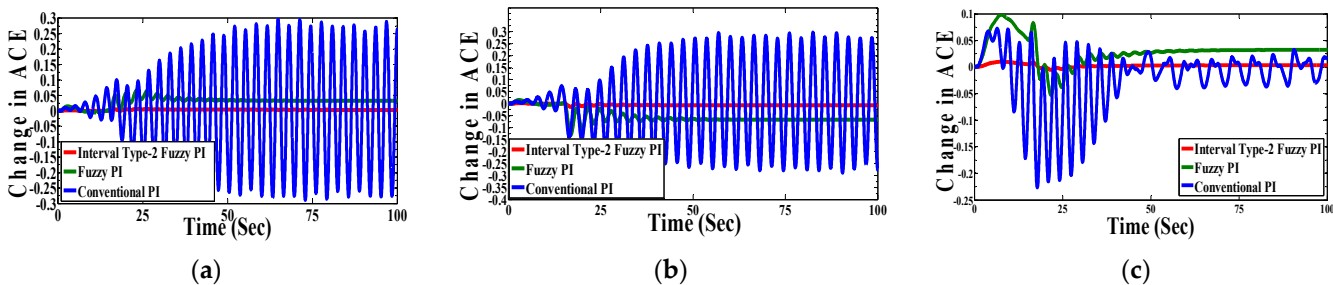

**Figure 20.** ACE variations of (**a**) area 1, (**b**) area 2 and (**c**) area 3 using IT2FLC fuzzy-PI, fuzzy-PI and PI controller due to step load change of 0.1 p.u. in area 2.

### 4.3. Condition 3

Figures 21–23 show the frequency variation, inter-tie-line power change and ACE, respectively, for the three different areas due to a 0.1 per unit (0.001 MW) change in step load demand in area 3.

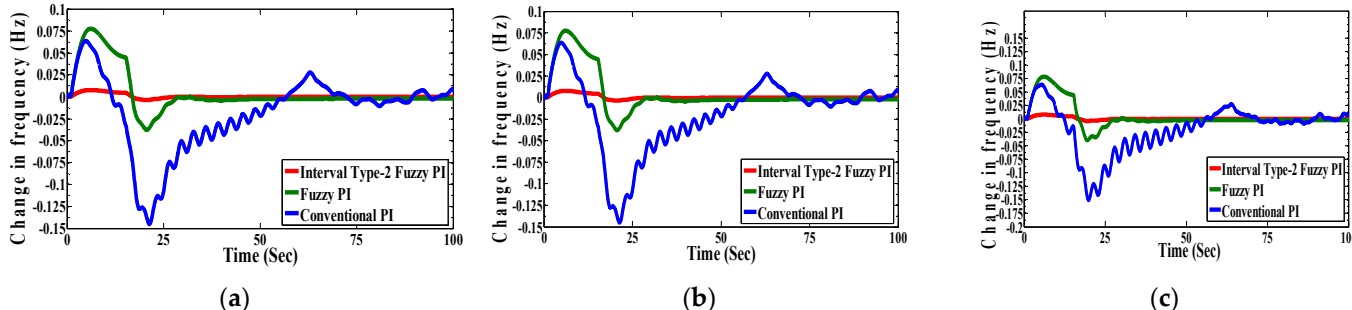

**Figure 21.** Frequency variations of (**a**) area 1, (**b**) area 2 and (**c**) area 3 using IT2FLC fuzzy-PI, fuzzy-PI and PI controller due to step load change of 0.1 p.u. in area 3.

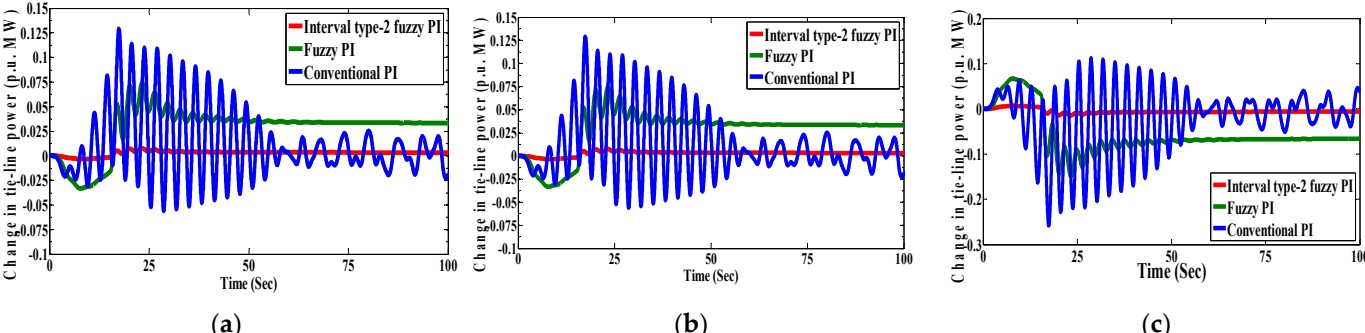

**Figure 22.** Tie-line power variations ($\Delta P_{tie}$) of (**a**) area 1, (**b**) area 2 and (**c**) area 3 using IT2FLC fuzzy-PI, fuzzy-PI and PI controller due to step load change of 0.1 p.u. in area 3.

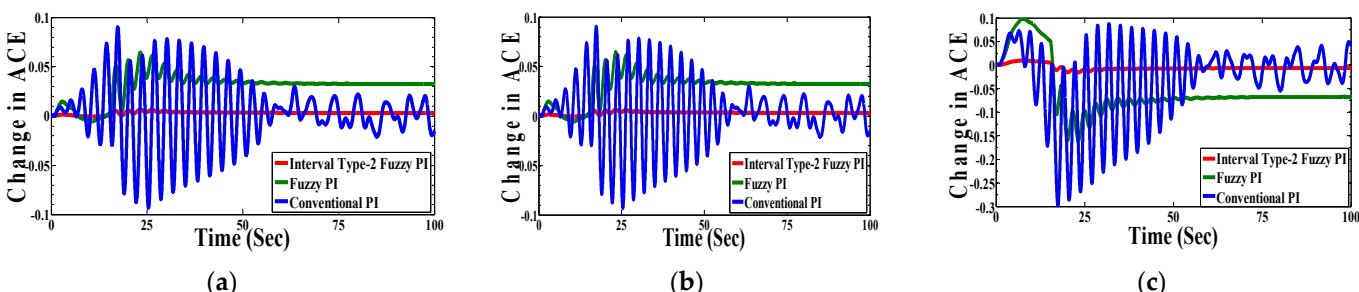

**Figure 23.** ACE variations of (**a**) area 1, (**b**) area 2 and (**c**) area 3 using IT2FLC fuzzy-PI, fuzzy-PI and PI controller due to step load change of 0.1 p.u. in area 3.

### 4.4. Condition 4

Figures 24–26 show the frequency variation, inter-tie-line power change and ACE, respectively, for the three different areas due to a 0.1 per unit (0.001MW) change in step load demand in all three areas.

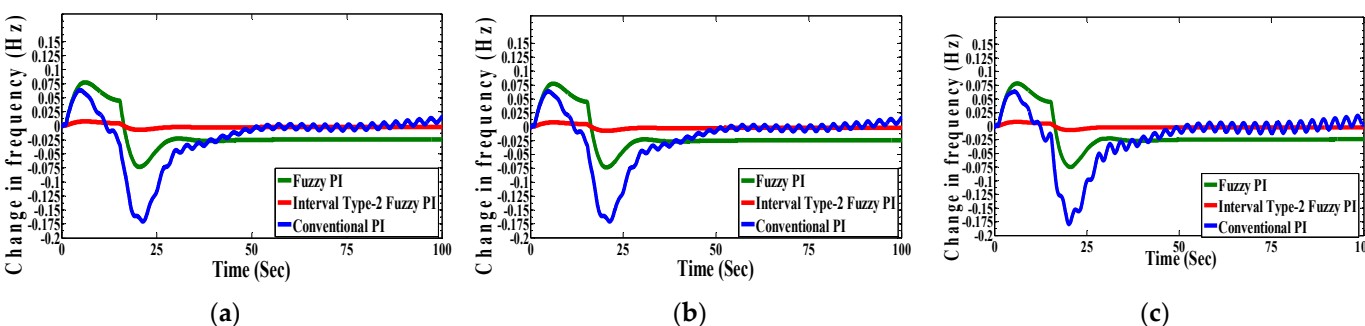

**Figure 24.** Frequency variations of (**a**) area 1, (**b**) area 2 and (**c**) area 3 using IT2FLC fuzzy-PI, fuzzy-PI and PI controller due to step load change of 0.1 p.u. in all three areas.

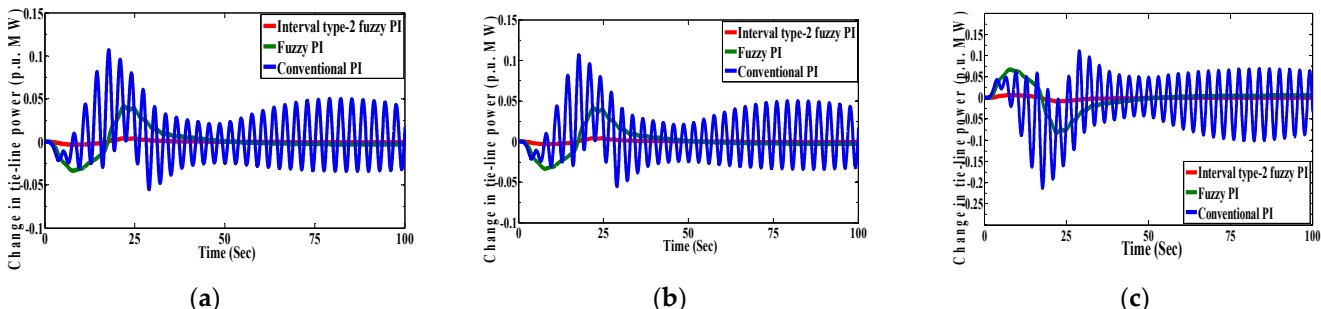

**Figure 25.** Tie-line power variations ($\Delta P_{tie}$) of (**a**) area 1, (**b**) area 2 and (**c**) area 3using IT2FLC fuzzy-PI, fuzzy-PI and PI controller due to step load change of 0.1 p.u. in all three areas.

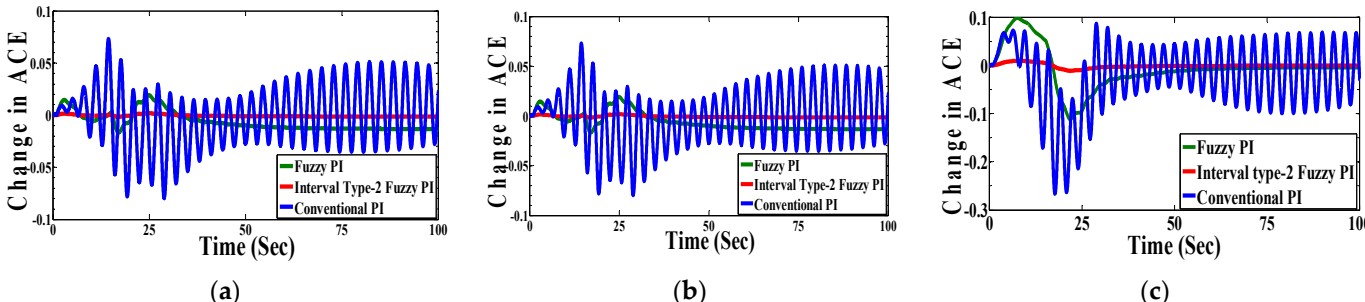

**Figure 26.** ACE variations of (**a**) area 1, (**b**) area 2 and (**c**) area 3 using IT2FLC fuzzy-PI, fuzzy-PI and PI controller due to step load change of 0.1 p.u. in all three areas.

An increment of 0.1 per unit (0.001 MW) step load in all areas applied under four different cases was investigated. It was found that frequency variation, tie-line power and ACE rapidly returned to zero using the type-2 fuzzy PI approach.

The performance for the above case study is illustrated numerically in Table 2. Itis obvious from Table 2 that type-2 fuzzy outperformed fuzzy PI and conventional PI approaches. In this table, $\Delta f_1$, $\Delta f_2$ and $\Delta f_3$ are changes in frequency; $\Delta ACE_1$, $\Delta ACE_2$ and $\Delta ACE_3$ are area control errors; $\Delta P_{tie1}$, $\Delta P_{tie2}$ and $\Delta P_{tie3}$ are tie-line power fluctuations in areas 1, 2 and 3, respectively.

**Table 2.** Performance analysis of various control approaches used in the test system.

| Case | Parameters | Controller | Settling Time (s) | Peak Time (s) | Maximum Overshoot |
|------|-----------|-----------|-------------------|---------------|-------------------|
| 1 | $\Delta f_1$ | PI | 68 | 6 | 0.056 |
| | | Fuzzy PI | 45 | 7.8 | 0.051 |
| | | IT-2 Fuzzy PI | 39 | 7 | 0.01 |
| | $\Delta f_2$ | PI | 62 | 6 | 0.054 |
| | | Fuzzy PI | 40 | 8 | 0.049 |
| | | IT-2 Fuzzy PI | 38 | 7.8 | 0.009 |
| | $\Delta f_3$ | PI | 60 | 7 | 0.054 |
| | | Fuzzy PI | 38 | 7.9 | 0.05 |
| | | IT-2 Fuzzy PI | 30 | 8 | 0.009 |
| | $\Delta ACE_1$ | PI | >100 | 3 | 0.025 |
| | | Fuzzy PI | 50 | 3.5 | 0.02 |
| | | IT-2 Fuzzy PI | 45 | 2 | 0.01 |
| | $\Delta ACE_2$ | PI | >100 | 3 | 0.026 |
| | | Fuzzy PI | 55 | 3.5 | 0.021 |
| | | IT-2 Fuzzy PI | 42 | 2.5 | 0.009 |
| | $\Delta ACE_3$ | PI | 59 | 4.7 | 0.1 |
| | | Fuzzy PI | 70 | 9.8 | 0.065 |
| | | IT-2 Fuzzy PI | 60 | 8 | 0.008 |
| | $\Delta P_{tie1}$ | PI | >100 | 10.5 | 0.01 |
| | | Fuzzy PI | 60 | 20 | 0 |
| | | IT-2 Fuzzy PI | 52 | 15 | 0.008 |
| | $\Delta P_{tie2}$ | PI | >100 | 10.4 | 0.012 |
| | | Fuzzy PI | 60.5 | 19 | 0 |
| | | IT-2 Fuzzy PI | 57 | 15 | 0.009 |
| | $\Delta P_{tie3}$ | PI | 60 | 4.8 | 0.04 |
| | | Fuzzy PI | 55 | 10 | 0.065 |
| | | IT-2 Fuzzy PI | 52 | 7 | 0.01 |
| 2 | $\Delta f_1$ | PI | 65 | 6 | 0.06 |
| | | Fuzzy PI | 48 | 7.8 | 0.075 |
| | | IT-2 Fuzzy PI | 42 | 6 | 0.008 |
| | $\Delta f_2$ | PI | 68 | 6 | 0.064 |
| | | Fuzzy PI | 47.5 | 7.8 | 0.075 |
| | | IT-2 Fuzzy PI | 39 | 6 | 0.01 |
| | $\Delta f_3$ | PI | 63 | 7 | 0.055 |
| | | Fuzzy PI | 48 | 8 | 0.074 |
| | | IT-2 Fuzzy PI | 35 | 5.6 | 0.012 |
| | $\Delta ACE_1$ | PI | >100 | 3 | 0.01 |
| | | Fuzzy PI | 55 | 3.6 | 0.012 |
| | | IT-2 Fuzzy PI | 43 | 2 | 0.009 |
| | $\Delta ACE_2$ | PI | >100 | 3 | 0.02 |
| | | Fuzzy PI | 49 | 3.5 | 0.022 |
| | | IT-2 Fuzzy PI | | | 0.01 |
| | $\Delta ACE_3$ | PI | 59 | 4.7 | 0.058 |
| | | Fuzzy PI | 68 | 9.8 | 0.1 |
| | | IT-2 Fuzzy PI | 48 | 6 | 0.013 |
| | $\Delta P_{tie1}$ | PI | >100 | 10.5 | 0.01 |
| | | Fuzzy PI | 55 | 22.2 | 0.052 |
| | | IT-2 Fuzzy PI | 46 | 18 | 0.008 |
| | $\Delta P_{tie2}$ | PI | >100 | 10.4 | 0.01 |
| | | Fuzzy PI | 66 | 20 | 0 |
| | | IT-2 Fuzzy PI | 50 | 15 | 0.007 |
| | $\Delta P_{tie3}$ | PI | 60 | 4.8 | 0.048 |
| | | Fuzzy PI | 70 | 10 | 0.058 |
| | | IT-2 Fuzzy PI | 58 | 7.1 | 0.007 |

**Table 2.** *Cont.*

| Case | Parameters | Controller | Settling Time (s) | Peak Time (s) | Maximum Overshoot |
|---|---|---|---|---|---|
| 3 | $\Delta f_1$ | PI | 83 | 6 | 0.06 |
| | | Fuzzy PI | 48 | 7.8 | 0.075 |
| | | IT-2 Fuzzy PI | 34 | 5.2 | 0.012 |
| | $\Delta f_2$ | PI | 83 | 6 | 0.061 |
| | | Fuzzy PI | 46 | 7.8 | 0.076 |
| | | IT-2 Fuzzy PI | 35 | 6 | 0.01 |
| | $\Delta f_3$ | PI | 96 | 7 | 0.06 |
| | | Fuzzy PI | 46 | 7.9 | 0.08 |
| | | IT-2 Fuzzy PI | 38 | 6.4 | 0.009 |
| | $\Delta ACE_1$ | PI | 73 | 2.7 | 0.0149 |
| | | Fuzzy PI | 51 | 3.5 | 0.018 |
| | | IT-2 Fuzzy PI | 43 | 2.6 | 0.005 |
| | $\Delta ACE_2$ | PI | 73 | 2.7 | 0.015 |
| | | Fuzzy PI | 50 | 3.5 | 0.016 |
| | | IT-2 Fuzzy PI | 42 | 2.3 | 0.007 |
| | $\Delta ACE_3$ | PI | 75 | 4.8 | 0.054 |
| | | Fuzzy PI | 58 | 9.8 | 0.1 |
| | | IT-2 Fuzzy PI | 43 | 7 | 0.013 |
| | $\Delta P_{tie1}$ | PI | 73 | 6.5 | 0.015 |
| | | Fuzzy PI | 47.3 | 22.2 | 0.05 |
| | | IT-2 Fuzzy PI | 34 | 14 | 0.011 |
| | $\Delta P_{tie2}$ | PI | 73 | 6.5 | 0.015 |
| | | Fuzzy PI | 51.5 | 22.2 | 0.058 |
| | | IT-2 Fuzzy PI | 42 | 16 | 0.013 |
| | $\Delta P_{tie3}$ | PI | 75.4 | 4.8 | 0.049 |
| | | Fuzzy PI | 62 | 10 | 0.054 |
| | | IT-2 Fuzzy PI | 53 | 7.4 | 0.008 |
| 4 | $\Delta f_1$ | PI | 72 | 6 | 0.061 |
| | | Fuzzy PI | 38 | 8 | 0.075 |
| | | IT-2 Fuzzy PI | 35 | 7.1 | 0.014 |
| | $\Delta f_2$ | PI | 72 | 6 | 0.062 |
| | | Fuzzy PI | 38 | 8 | 0.076 |
| | | IT-2 Fuzzy PI | | | 0.013 |
| | $\Delta f_3$ | PI | 69 | 7 | 0.065 |
| | | Fuzzy PI | 40 | 8 | 0.075 |
| | | IT-2 Fuzzy PI | 37 | 6.2 | 0.016 |
| | $\Delta ACE_1$ | PI | >100 | 2.6 | 0.014 |
| | | Fuzzy PI | 82 | 3.5 | 0.018 |
| | | IT-2 Fuzzy PI | 60 | 2 | 0.007 |
| | $\Delta ACE_2$ | PI | >100 | 2.6 | 0.013 |
| | | Fuzzy PI | 82 | 3.5 | 0.016 |
| | | IT-2 Fuzzy PI | 60 | 2 | 0.007 |
| | $\Delta ACE_3$ | PI | >100 | 4.8 | 0.058 |
| | | Fuzzy PI | 70 | 10 | 0.1 |
| | | IT-2 Fuzzy PI | 64 | 9 | 0.011 |
| | $\Delta P_{tie1}$ | PI | >100 | 6.5 | 0.015 |
| | | Fuzzy PI | 72 | 28.8 | 0.04 |
| | | IT-2 Fuzzy PI | 65 | 23 | 0.01 |
| | $\Delta P_{tie2}$ | PI | >100 | 6.5 | 0.014 |
| | | Fuzzy PI | 72 | 28.8 | 0.038 |
| | | IT-2 Fuzzy PI | 65 | 23 | 0.009 |
| | $\Delta P_{tie3}$ | PI | >100 | 4.7 | 0.048 |
| | | Fuzzy PI | 75 | 10 | 0.058 |
| | | IT-2 Fuzzy PI | 67 | 6 | 0.008 |

## 5. Conclusions

This paper presents a design of interval type-2 fuzzy-based, dual-mode gain scheduling of a PI controller for load frequency control in nonlinear, three-area power systems considering RESs. It was observed that the proposed controller outperformeda conventional PI controller and a type-I fuzzy PI controller in terms of frequency minimization, stability and robustness of the test system. For different load conditions, this approach was found suitable not only to control frequency variation, but also to minimize tie-line power fluctuation and reduce the change in ACE. Further, to reveal the performance of the proposed method, $t_s$ (settling time), $M_p$ (maximum overshoot) and $t_p$ (peak time), were tabulated. In this way, the design of a IT2FLC-based DMGS of PI controller was validated.

**Author Contributions:** Conceptualization, B.J., S.R. and M.K.P.; methodology, S.R., B.J. and M.K.P.; software, S.R., B.J. and M.K.P.; validation, B.J., M.K.P. and S.R.; formal analysis, B.J.; investigation, J.K.; resources, S.R., B.J., M.K.P. and J.K.; writing—S.R.; writing—review and editing, B.J. and M.K.P.; visualization, J.K.; supervision, B.J. and M.K.P.; project administration, B.J. and M.K.P. All authors have read and agreed to the published version of the manuscript.

**Funding:** The Article Processing Charge (APC) will be paid by authors.

**Data Availability Statement:** Not applicable.

**Acknowledgments:** The contribution of research scholars of the department is greatly acknowledged.

**Conflicts of Interest:** The authors declare that there is no conflict of interest.

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
