# Peer review of "Interval Type-2 Fuzzy Logic Control-Based Frequency Control of Hybrid Power System Using DMGS of PI Controller"

_applsci, doi:10.3390/app112110217_

Round 1

Reviewer 1 Report

This paper proposes a hybrid micro grid power system in which non-conventional energy sources are added to each area of the conventional power plant which makes the system much more prone to frequency variations. Moreover, an interval type-2 fuzzy logic-based dual-mode gain scheduling (DMGS) of the proportional and integral controller is presented. Experiments verify the effectiveness of the proposed approach over simple fuzzy PI and conventional PN controlling approach.

In my opinion, this is well organized and written, and includes some interesting contents. But, I have some major concerns and comments regarding the technical and writing aspects.

  1. A variety of abbreviations are not clearly defined throughout this paper, such as IPD, FLC, PI, etc. Please go through this paper and elaborate.
  2. The abstract is a little bit lengthy and thus could be shorten properly, only most important contributions and insights are included in the abstract.
  3. The quality of some figures, e.g., Figs. 1, 3, 4, 7, 26, are too low to be identified. All figures should be re-plotted so as to make all items clear.
  4. The authors should consider more baselines in the state-of-the-art literature so as to substantially verify the effectiveness of the proposed schemes.
  5. A lot of mathematic expressions and formulas are not formatted normatively, it seems quite casual to make it, such as the equations (8-9), and the notations between them.
  6. Some related works are missed, especially in the recent years, such as

[a] Z. Ma, M. Xiao, Y. Xiao, Z. Pang, H. V. Poor, and B. Vucetic, “High-reliability and low-latency wireless communication for internet of things: Challenges, fundamentals, and enabling technologies,” IEEE Internet Things J., vol. 6, no. 5, pp. 7946–7970, 2019.

[b] Protograph LDPC-coded BICM-ID with irregular CSK mapping in visible light communication systems, IEEE Transactions on Vehicular Technology, 2021, 70(10): 1-6, DOI 10.1109/TVT.2021.3106053

Author Response

Please see the attachment. Thank you for your significant comments. These comments are really appreciable. It will help authors to enhance the quality of the paper.

Reviewer 2 Report

Authors claimed to have the frequency control of micro grid using interval type-2 fuzzy control based dual mode gain scheduling of PI controller. The article is well written. However, it needs some minor modifications.

The following are some concerns:
Comment 1: Why have you used type-2 fuzzy logic technique in proposed test system?
Comment 2: Significance of your work where this work can be deployed?
Comment 3: What is importance of dual mode gain scheduling technique with controller?
Comment 4: How does GRC affect system stability?
Comment 5: The literatures written in the introduction should be elaborated.
Comment 6: The importance of dynamic selector switch used in this article. Mention it.
Comment 7: Why the load changing is kept same in all the four cases.

Author Response

Please see the attachment. The reviewer's comments are very significant and helpful for improvement in manuscript.

Round 2

Reviewer 1 Report

The authors have carefully revised the paper and addressed all my comments. I can find that the revision is of good shape. Now, it is ready for publication. No more comments.